# Neonatal Tracheal Intubation in the NICU: A Literature Review

**DOI:** 10.3390/healthcare13111242

**Published:** 2025-05-24

**Authors:** Jenna L. Schaefer-Randolph, Spencer G. Shumway, Colin W. Fuller, Vikram G. Ramjee, Nilesh R. Vasan

**Affiliations:** 1College of Medicine, University of Oklahoma Health Sciences Center, Oklahoma City, OK 73104, USA; jenna-schaefer@ouhsc.edu (J.L.S.-R.); spencer-shumway@ouhsc.edu (S.G.S.); 2Department of Otolaryngology-Head and Neck Surgery, University of Oklahoma Health Sciences Center, Oklahoma City, OK 73104, USA; colin-fuller@ouhsc.edu (C.W.F.); vikram.ramjee@ouhealth.com (V.G.R.)

**Keywords:** pediatric intubation, newborn intubation, preterm intubation, neonatal resuscitation, neonatal airway management, video laryngoscopy, neonatal intubation outcomes, pediatric otorhinolaryngology

## Abstract

This literature review explores factors influencing neonatal ICU intubation success, compares outcomes across settings, and identifies strategies to improve outcomes in this vulnerable population. A PubMed search was conducted using relevant keywords related to neonatal tracheal intubation. Studies published in English from 2000 to 2024 were included, with additional sources identified through manual bibliography reviews. Extracted findings were qualitatively synthesized by themes such as procedural outcomes, intubation setting, and provider training level. Nearly half of all neonatal tracheal intubations (TIs) are unsuccessful, with the rate of tracheal intubation adverse events (TIAEs) increasing with each additional attempt. First-pass success rates (FPSRs) correlate with provider experience, with attending physicians achieving the highest rates across all settings. Video laryngoscopy is associated with improved outcomes, particularly in neonates with difficult airways; however, direct laryngoscopy (DL) remains more commonly used. Premedication has been shown to reduce TIAEs and enhance FPSR, yet it remains underutilized in clinical practice. Standardized protocols, improved simulation-based training, and multidisciplinary strategies are essential to reduce complications. Future research should prioritize optimizing airway management and evaluating the impact of otorhinolaryngologist involvement, especially in difficult airway cases.

## 1. Introduction

Pediatric tracheal intubation (TI) is a critical, life-saving procedure that provides essential respiratory support for young children unable to breathe adequately on their own. Despite its importance, neonatal intubation is dangerous, with the rate of neonatal tracheal intubation adverse events (TIAEs) ranging from 9% to 50%, underscoring the complexity and difficulty of the procedure [1]. Successful intubation in neonates is particularly critical due to their increased vulnerability, stemming from their small size, unique anatomy, and physiological characteristics. Complications such as hypoxemia, severe oxygen desaturation, and bradycardia are common, and the low first-pass success rate (FPSR) predisposes neonates to multiple intubation attempts [2].

Repeated intubation attempts are associated with development of severe TIAEs, including intraventricular hemorrhage, pneumothorax, direct airway trauma, cardiorespiratory arrest, and even death [3]. Achieving success in neonatal TI requires thorough preparation, vigilant monitoring, specialized equipment, and highly experienced operators. Despite these measures, a difficult airway can still occur, characterized by challenges that exceed the usual in achieving secure and effective placement of an endotracheal tube (ETT) in the neonate’s trachea [4]. While most neonatal TIs are performed by neonatologists, pediatric intensivists, and anesthesiologists, intervention from an otolaryngologist (ENT) may be necessary to achieve adequate ventilation, especially when a difficult airway is suspected.

This literature review aims to elucidate the factors affecting intubation success rates in neonatal ICUs, compare these rates across different settings, and identify strategies to improve intubation success in neonates. In doing this, we employed a narrative review focused on specific challenges and innovations of TI related to neonates. Given the heterogeneity and expansiveness of the literature on TI, a comprehensive systematic review was not feasible. By examining these aspects, we hope to enhance understanding and practices concerning neonatal tracheal intubation, ultimately improving outcomes for this vulnerable population.

## 2. Materials and Methods

### 2.1. Search Strategy

A comprehensive literature search was conducted using PubMed to identify relevant studies published on neonatal intubation. A cursory sweep of other search engines (Embase, Cochrane Library, Scopus, CINAHL, and Web of Science) was conducted, but frequent overlap exists among high-impact studies, and most studies were ultimately found using PubMed. The primary keywords and Medical Subject Headings (MeSH) included “neonatal intubation”, “tracheal intubation”, “neonatal TI success rates”, “neonatal TIAE”, “neonatal intubation techniques”, “neonatal TI outcomes”, “pediatric TI”, “neonatal airway management”, and “video laryngoscopy”. Boolean operators (AND/OR) were used to refine search results. Articles published in English from 2000 to 2024 were considered. Additional references were identified through manual searches of the bibliographies of relevant articles. Given the narrative nature of this review, we focused on high-impact or frequently cited studies. We did not aim for comprehensive inclusion or a systematic review, but rather thematic breadth.

### 2.2. Inclusion and Exclusion Criteria

Studies were included if they:-Addressed neonatal intubation.-Provided original research, reviews, or clinical guidelines.-Were published in peer-reviewed journals.

Studies were excluded if they:-Did not focus on neonatal intubation techniques or outcomes.-Were case reports, opinion pieces, or commentaries without data.-Involved animal studies or non-human data.

### 2.3. Data Extraction and Analysis

Key information from selected articles was extracted, including study objectives, methods, findings, and conclusions. The findings were qualitatively synthesized and grouped based on emerging themes in neonatal intubation, such as patient outcomes, location of intubation, performer characteristics, and procedure challenges. No formal statistical analysis was conducted due to the narrative nature of this literature review.

## 3. Review of Literature

### 3.1. Neonatal Intubation: Indications and Importance

#### 3.1.1. Definition of and Indications for Neonatal Intubation

Neonatal tracheal intubation is a critical, life-saving procedure used to secure an airway in newborns who cannot maintain adequate ventilation independently. It involves inserting a tube through the mouth (orotracheal) or nose (nasotracheal) into the trachea, which is then connected to a mechanical ventilator or manual resuscitation bag to assist or control the patient’s breathing efforts. As obligate nasal breathers, nasal intubation of neonates can lead to obstruction of the nasal passages, and therefore orotracheal intubation is preferred [2,5]. A summary of common indications for neonatal intubation is given below in Table 1.

#### 3.1.2. Importance of Successful Intubation

Nearly half of all endotracheal intubations in neonates are unsuccessful, with the FPSR ranging from 30% to 57% [2]. Achieving intubation on the first attempt is crucial, as the rate of TIAEs strongly correlates with the number of TI attempts, ranging from 3.6% for a successful first attempt to 35% for two or more attempts [1,8,10]. FPS reduces the risk of oxygen desaturation, bradycardia, and hypoxemia, which can have devastating impacts on neonates, especially when compared with older populations: findings from two studies suggest that TIAEs occurred in 40% of neonatal intubations versus 20% among children [11,12]. Additionally, improper techniques can lead to airway trauma, including bleeding or swelling, further complicating subsequent intubation attempts and increasing the risk of chronic lung disease [7].

Neonates are particularly vulnerable to complications sustained during tracheal intubation, given their underdeveloped respiratory, cardiovascular, and neurological systems [2,6,13]. These factors heighten their susceptibility to experiencing severe complications such as hypoxia, bradycardia, and airway trauma during intubation, potentially leading to detrimental long-term impacts, including chronic lung disease and neurodevelopmental issues [7,14]. Moreover, TIAEs may disrupt critical neurological development, potentially resulting in long-term cognitive and neurodevelopmental deficits [14]. In fact, even with FPS, the rate of significant neurological impairment (SNI) and/or death still nears 20% [14]. The challenges in recognizing and managing these events in neonates, compounded by their inability to communicate symptoms, further underscore the need for meticulous technique and vigilant monitoring to minimize adverse outcomes. Ultimately, ensuring a high FPSR is critical in providing safe and effective respiratory support in this vulnerable population.

### 3.2. Settings for Tracheal Intubation

Most neonatal TIs are performed in the NICU setting, which allows the procedure to occur in a controlled manner in proximity to specialized providers and equipment. In a 2020 retrospective study on neonatal TIs, 73% occurred in the NICU setting, whereas 27% were intubated in the delivery room (DR) [15]. This is unsurprising, as many neonates in the NICU have conditions that predispose them to respiratory failure, such as prematurity, congenital anomalies, or sepsis [8]. Ultimately, these conditions often necessitate intubation for adequate respiratory support. However, TIAE incidence and severity is similar regardless of the setting in which the TI is performed [8,15].

#### 3.2.1. Neonatal Intensive Care Unit (NICU)

While rates of TIAE may be similar across settings, several studies suggest the NICU boasts the highest FPSR, ranging from 37% to 54%; however, rates are lower in patients with difficult airways and are dependent on provider training or experience level [9,16,17]. Several studies have found a strong correlation between provider training level and neonatal TI success within two attempts, with attending physicians having the highest success rate across all settings [1,16]. In the NICU setting, the majority of first-attempt intubations are attempted by pediatric and neonatal attendings [8,18] and advanced practice practitioners [1]. According to an international NEAR4NEOS study of 2608 pediatric primary TIs in the NICU setting, the FPSRs for attending, fellow, and resident were 60%, 53%, and 23%, respectively [18]. Similar observations have been recorded in the PICU setting [19].

Surprisingly, no correlation was discovered between number of TIAEs and provider training level, although oxygen desaturation occurred in nearly 50% of TIs, and most frequently when performed by resident physicians [16,18]. It is important to note that a recent decline in intubation opportunities for residents has been described, which may be due to limits on resident working hours [20], increased presence of advanced practice practitioners in NICUs, and a shift away from invasive neonatal intubation practices [1,17]. This decline was documented in a retrospective cohort study of 25 PICUs in the US, which examined pediatric resident TI attempts between 2010 and 2016. Their results suggested an annual 3.4% decline in resident laryngoscopy participation [21]. Furthermore, a multicenter retrospective study spanning 2014–2017 found that out of 2009 intubations in the NICU setting, the majority were performed by advanced practice providers (38%), followed by fellows (30%) and residents (15%) [1]. Most recently, the COVID-19 pandemic significantly worsened the already limited pediatric intubation opportunities for residents. A study by Miller et al. compared pre-pandemic data from January 2017 to March 2020 with those from March 2020 to March 2021, during the pandemic. The study found that pediatric resident intubation attempts decreased nearly sixfold, along with a decline in attempts performed by pediatric emergency medicine fellows [22]. Ultimately, a balance must be established to minimize procedural risk while ensuring pediatric resident competencies are achieved.

#### 3.2.2. Delivery Room (DR)

The delivery room is the second-most common setting for neonatal TI [2]. However, due to lung fragility, surfactant deficiency, and weak respiratory drive, preterm infants and neonates born at very low birth weight are most at risk of requiring DR intubation, and also are most vulnerable to lung-associated TIAEs such as bronchopulmonary dysplasia [23,24]. The literature estimates that between 3% and 8% of newborns require airway support within the first few minutes after birth; however, there has been a recent shift towards non-invasive support, especially for premature neonates [23,25]. A study examining infants born between 24 and 27 gestational weeks found an increased risk of neurodevelopmental impairment at three years for those intubated in the delivery room [26]. Another study involving infants born between 23 and 32 weeks concluded that while the rate of SNI or death was as low as 2.5% among non-intubated infants, this risk rose to 12% for those intubated in the NICU and 32% for those intubated in the DR [14].

While the overall first-attempt success rate in this setting is 46%, attending neonatologists had the highest success rate of all providers (64%), followed by neonatology fellows (52%), who perform most DR intubations [1]. While less common than in the NICU setting, nearly a third of neonatal intubation attempts will result in oxygen desaturation [1,16]. This variance may be explained by differences in clinical environment and patient population, as well as the complexity of health conditions (prematurity, respiratory distress syndrome, or congenital anomalies), which can increase desaturation risk, especially amongst critically ill neonates.

#### 3.2.3. Operating Room (OR)

In the United States, elective admissions account for 40% of pediatric surgical operating room (OR) visits [27]. However, critically ill neonates in the NICU, who frequently require surgical procedures, represent a much smaller portion of these elective admissions compared to the general pediatric population. This is largely because many neonatal surgeries are emergent rather than elective and often require TI. In contrast, when a TI is planned for elective surgery, patients typically do not show adverse indications like poor respiratory drive or hypoxia.

Moreover, in the OR, TI usually takes place after anesthesia induction, providing a controlled environment that contributes to heightened safety [28]. As a result, intubating pediatric patients in the OR is often considered a safer approach, rendering pediatric patients undergoing elective surgeries ideal candidates for TI training. Supporting this, Riva et al. conducted a randomized controlled trial involving neonates and infants scheduled for elective TI in the OR. Their study found that using video laryngoscopes (VLs) rather than direct laryngoscopes (DLs) further improved success rates [29].

However, not all intubations in the OR occur under optimal conditions. Complications like hypoxemia remain a significant concern, as it frequently arises during airway procedures and TI attempts in pediatric patients. According to the National Emergency Airway Registry for Children (NEAR4KIDS), the overall hypoxemia rate is 13% for all intubations and nearly 50% for those classified as difficult intubations [30,31]. This risk is particularly pronounced in infants and neonates, who are more susceptible to hypoxemia due to their higher metabolic requirements and increased likelihood of alveolar collapse during general anesthesia [32].

For neonates with particularly difficult anatomy, other techniques can be utilized in the OR to minimize TIAEs. Sarkar et al. described a case report where a Hopkins rod was used by an otolaryngologist to visualize the airway, while an anesthesiologist successfully intubated using a Macintosh 2 blade [33]. It is the authors’ own experience that Hopkins rods provide a clear and magnified view of the airway, helping to navigate difficult airway situations, treat airway defects, and foreign body removal. An ETT can be inserted over a Hopkins rod bronchoscope to facilitate intubation in complex airway cases, a technique often employed by pediatric otolaryngologists. The use of Hopkins rods is typically limited to controlled environments, such as the NICU or operating room [34]. An example of this type of equipment can be seen in Figure 1.

### 3.3. Challenges in Neonatal Intubation

Neonatal intubation presents numerous challenges, particularly in difficult airway situations, defined by failure to secure ventilation with ETT, laryngoscopy, or a supraglottic device [4]. Due to complex anatomical and physiological characteristics, difficult airways amongst neonates are common, and occur in 14% of intubations performed in the NICU; however, premature (<32 weeks’ gestation) and low birth weight (<1500 g) neonates are most at risk and can be further complicated by congenital or acquired abnormalities [3,4]. As neonates are particularly vulnerable to hypoxia, these situations may necessitate a specialist with the highest level of expertise, such as an otolaryngologist who specializes in pediatric or neonatal airway management [35].

#### 3.3.1. Anatomical Challenges

Neonatal anatomy significantly differs from that of older children and adults and contributes to the lower success rates of tracheal intubations in newborns. Neonates have a proportionally larger tongue, smaller and more flexible airway, and a higher and more anterior larynx, making visualization of the vocal cords more challenging [6]. The trachea is also shorter and narrower, increasing the risk of misplaced intubation, such as into the right mainstem bronchus, or accidental extubation [36,37]. Additionally, the neonatal epiglottis is relatively longer and floppier, which can obstruct the view during intubation attempts [3,6,36,37]. Furthermore, neonates possess more flexible cartilage in their airways, making them more susceptible to collapse and complicating the maintenance of an open airway during the procedure [36]. While this cartilage flexibility may enhance the effectiveness of anterior cricoid pressure, which has been shown to effectively close the esophageal entrance during intubation, the overall fragility and smaller size of neonates make tracheal intubation more complex and contribute to its lower success rates compared to adults [38]. An example of a smaller size of laryngoscope, designed for smaller airways, is shown in Figure 2.

#### 3.3.2. Physiological Challenges

Neonatal physiology is also a significant contributor to the lower success rates of tracheal intubations in newborns. At birth, neonates experience a complex transition from placental gas exchange to breathing through their lungs, shifting from fetal to neonatal circulation. This transition increases the arterial partial pressure of oxygen and reduces pulmonary vascular resistance, which is essential for the closure of fetal shunts like the ductus arteriosus and foramen ovale. Successful closure of these shunts allows blood to circulate properly through the lungs and systemic circulation. However, any instability during this process, such as difficulties in oxygenation or ventilation, can disrupt this transition. This may lead to desaturation, elevated pulmonary pressures, and impaired shunt closure, resulting in persistent fetal circulation. This condition, characterized by sustained right-to-left shunting, can cause severe complications, including hypoxemia and bradycardia [39].

Compared to adults, neonates also have a higher metabolic rate and increased oxygen consumption, leading to rapid desaturation during intubation attempts [13]. Their immature respiratory muscles, smaller lung volumes, and limited functional residual capacity also make it difficult to maintain adequate ventilation and oxygenation, increasing both the urgency and difficulty of the procedure [6]. Additionally, neonates’ autonomic nervous systems are less developed and they have a lower threshold for hypoxia, making them more prone to hypoxia and bradycardia conditions that can quickly become life-threatening with prolonged or repeated intubation attempts [2,13].

The airway tissues in neonates are also delicate and reactive, making them more susceptible to trauma and swelling, which can further complicate intubation by increasing resistance and potentially causing airway obstruction [7]. These physiological challenges make neonates particularly vulnerable to intubation failures and complications, underscoring the need for a high level of skill and precision during tracheal intubations in this population.

### 3.4. Reintubation in the NICU

Prolonged intubation in the neonatal population is associated with adverse neurodevelopmental and respiratory outcomes, such as bronchopulmonary dysplasia (BPD). Therefore, early extubation is ideal. However, if a neonate exhibits signs of inadequate breathing and oxygenation after extubation, reintubation may be necessary. According to a 2018 multicenter observational study, 48% of neonates required at least one reintubation following elective extubation, with the risk of BPD and death significantly increasing if reintubation occurred within 48 h [40]. While reintubation rates vary significantly across studies, common reasons for reintubation include unplanned extubation followed by planned extubation failure (EF), with reintubation rates of 58% and 12%, respectively [41].

#### 3.4.1. Extubation Failure

Extubation failure (EF) is defined as necessary reintubation within 2–7 days of extubation and occurs when a neonate cannot sustain adequate breathing after the initial, planned removal of the endotracheal tube. This can result from muscle fatigue, insufficient respiratory drive, upper airway obstruction, or unresolved medical conditions. Unfortunately, EF is not uncommon and is associated with additional long-term complications such as increased BPD severity, longer hospital stays, and death [40,42]. Premature infants are perhaps most at risk of EF, with one study estimating a 24% failure rate within 5 days [43]. This outcome was paralleled by a large cohort study on 1348 low birth weight (<2500 g) neonates, which estimated a 26% EF rate [44]. While EF is not uncommon in the NICU, premature neonates are most at risk, with the strongest predictors being lower gestational age and birth weight [45].

#### 3.4.2. Unplanned Extubation

Unplanned extubation (UE) refers to the accidental or inadvertent removal of an endotracheal tube from a patient who requires mechanical ventilation. Compared to other patient populations, neonates are most at risk of UE, with estimates ranging from 0.14 to 5.3 UEs per 100 patient intubated days [46], with 86% requiring reintubation within the hour [47]. It is important to note that across studies, there is immense variation in UAE rates, likely due to differences in size, level of care, and protocols within NICUs. UE is the fourth most common preventable adverse event in the NICU setting, with common causes including infant movement, inadequate securing of the tube, or handling during medical procedures. Krishnan et. al. estimated that 50% of UEs were caused by loose tape, inadequate sedation, and retaping. Interestingly, initiatives to implement a standardized taping method resulted in a 48% reduction in UE rates, a finding consistent within the literature [47].

### 3.5. Strategies to Improve Intubation Success Rates in the NICU

Improving neonatal intubation success rates in the NICU requires a multifaceted approach that includes specialized training, optimized equipment and technique, and standardized protocols.

#### 3.5.1. Training and Simulation

Regular training and simulation are indispensable for enhancing the skills of healthcare providers, particularly in neonatal care. Based on adult literature suggesting a minimum of 40 successful intubations prior to achieving competency, anesthesiology residents have dedicated time built into their curriculum to achieve proficiency in intubation [17,48]. However, opportunities for practice in pediatric populations are scarce. This scarcity stems not only from the infrequent occurrence of neonatal intubations but also from the growing preference for less invasive intubation methods, the rising utilization of advanced practice providers (APPs), and the constraints imposed by regulations on resident working hours. Consequently, there has been a noticeable decline in intubation success rates, particularly among pediatric residents [18]. These factors contribute to skill fatigue and increase patient risk, underscoring the urgent need for enhanced training methodologies.

Addressing these challenges is crucial, given recent findings. A 2018 study demonstrated that implementing a standardized training program for neonatal intubations significantly improved the FPSR among junior trainees, increasing it from 37% to 61% and reducing the number of intubation attempts required [49]. While simulation offers a safe environment for trainees to practice and enhance their FPSR without patient risk, it is important to recognize that current manikin training does not fully simulate the complexities and stressors of real clinical scenarios.

#### 3.5.2. Equipment Innovations

Over the last few decades, several technological advances have significantly improved outcomes in the NICU setting, particularly in tracheal intubation. Traditionally, the Miller DL has been regarded as the gold-standard approach to TIs due to its straight, flat design, which contrasts with the curved shape of the Macintosh blade. The Miller blade is especially advantageous for infants and small children because it allows for better lifting of the epiglottis, offering a clearer view of the vocal cords and the glottis during intubation [50].

Unlike the traditional DL blades like the Miller and Macintosh, VLs incorporate a miniature camera at the blade’s tip, capturing high-resolution images of the airway and displaying them on a screen (Figure 3). This technology enhances visualization by magnifying and clarifying airway structures, including the vocal cords, aiding clinicians in challenging intubation scenarios where direct line of sight is limited. The adjustable camera angle also accommodates diverse patient anatomies and reduced neck mobility, enhancing safety and reducing trauma during intubation.

Although VL shows promise in training, its clinical advantages over DL are notable, especially in improving visualization and accuracy during intubation, which is crucial given the fragility of the neonatal airway [51,52]. One study found that when VL was used with standard blades and supplemental oxygen, it resulted in a higher FPSR in neonatal intubation compared to similar methods using DL, though there was no difference in hypoxemia rates [29,53]. Additionally, a randomized controlled trial involving 226 neonates in a single center found that those who underwent emergency TI in the NICU and DR with VL had a higher FPSR than those intubated with DL [54]. A multicenter RCT supported these findings, noting that VL also reduced TIAEs in anesthetized neonates, with particular efficacy in low-birth-weight and difficult airway cases [55].

Despite these benefits, DL remains predominant, with only 23.2% of NICU TIs utilizing VL [50,56]. This may be due to the bulkiness of even pediatric-specific VLs or the size of their blades, which can be too large for very small or premature neonates. The delicate nature of their airways necessitates extremely small, precise instruments, but given the low market demand compared to adult counterparts, such tools are often not readily available [57]. Additionally, VLs come with an increased cost and widespread adoption is not always feasible in certain areas.

Interestingly, recent studies have reported an increased use of VL in North America, particularly among patients with difficult airways [22,58]. A similar trend is emerging in the United Kingdom, where new guidelines jointly established by the *European Society of Anesthesiology and Intensive Care* and the *British Journal of Anesthesia* recommend VL with an age-adapted standard blade as the first choice for intubating neonates and infants [59]. Furthermore, a 2019 international study spanning Asia, North America, and Europe concluded that intubations using VL and paralytic premedication resulted in fewer adverse events in the NICU [1].

Advancements in miniature devices and fiberscopes have greatly improved neonatal intubation by enhancing maneuverability, control, and visualization. Fiberscopes equipped with tiny, flexible fiber-optic cables offer detailed, real-time imaging of the neonatal airway. This technology allows clinicians to see around corners and navigate complex anatomical structures with precision, ensuring safer and more accurate placement of the endotracheal tube. Supraglottic airways (SGAs) can also serve as a conduit during fiber-optic intubation, guiding less experienced clinicians through difficult airways while providing continuous oxygen delivery and ventilation, thereby reducing the risk of hypoxemia—a particular vulnerability in neonates [60]. A nonrandomized study found that in subjects under one year old, fiber-optic intubation via a supraglottic airway was more successful on the first attempt than VL, with success rates of 54% versus 36%, respectively. Although complication rates were similar between the two groups, the incidence of hypoxemia was lower when continuous ventilation through the supraglottic airway was maintained during the fiber-optic intubation attempt [61]. Fiberscopes also have recently been evaluated in cases of mild respiratory distress syndrome in neonates, facilitating the delivery of surfactant without laryngoscopy or anesthesia [62].

#### 3.5.3. Protocols and Guidelines

##### Premedication Guidelines

Neonatal airway management, especially in emergency situations, requires a well-coordinated approach due to significant anatomical and physiological differences from adults. The use of hospital-specific standardized protocols and checklists can help ensure consistency and reduce variability in techniques. One key area of concern is premedication, which is often used to prevent bradycardia during TI. Although premedication has been shown to double neonatal TI success rates, it is still considered optional in many hospitals, leading to inconsistent dosing and timing of administration [63]. Nowhere are inconsistencies in managing difficult airways more apparent than in the use of premedication. In 2010, the American Academy of Pediatrics recommended premedication for all nonemergent neonatal TIs. However, premedication continues to be omitted in a significant percentage of cases.

This lack of standardization can affect outcomes, but research has demonstrated that implementing consistent premedication practices for nonemergent TIs can significantly improve patient results. For instance, Shay et al. observed better outcomes in two academic-affiliated NICUs in the United States after standardizing premedication protocols [64]. A 2019 NEAR4NEOS study across 11 academic NICUs found that premedication was not used in 36% of intubation attempts. This aligns with other studies showing that despite strong evidence linking premedication—specifically sedation and neuromuscular blockade (NMBA)—to fewer TIAEs and higher FPSRs, it is still underutilized [65,66]. An American Academy of Pediatrics (AAP) clinical report recommends premedication for all nonemergent neonatal TIs, with preference for medications that have rapid onset and short duration of action [23,25]. The recommendation from the AAP was to utilize an analgesic agent of anesthetic dose of a hypnotic drug with strong considerations of a vagolytic and paralytic [23,25]. Concerns about neurotoxicity, potential suppression of spontaneous respiratory drive, and resistance to new protocols may explain the reluctance to fully adopt these practices.

As such, in hospitals with low premedication adoption rates, targeted education on its safety and efficacy should be prioritized. Educational interventions should address misconceptions about neurotoxicity and highlight evidence demonstrating improved outcomes. In addition, increasing adherence to premedication guidelines may require a multifaceted approach: embedding premedication checklists into electronic health records, offering simulation-based training for neonatal airway teams, conducting regular audits with feedback, and engaging local experts and senior clinicians to reinforce change in practice. These steps will not only foster heightened provider confidence but also help shift institutional culture towards consistent, evidence-based practice.

##### Rapid-Sequence Intubation (RSI)

Rapid-sequence intubation (RSI) is a widely used technique for tracheal intubation (TI) that aims to minimize TIAEs using specific medications such as atropine, sedatives, and neuromuscular blocking agents (NMBAs). Along with medication, RSI typically involves preoxygenation for 3–5 min and airway suctioning to extend the safe apneic period and reduce the risk of oxygen desaturation. However, certain factors, including the absence of intravenous access or abnormal airway anatomy, may limit the use of RSI in some cases.

RSI is commonly implemented worldwide. For example, in a multicenter study from Japan, 69% of pediatric emergency department (PED) intubations used RSI protocols, with NMBA premedication administered in 69% of PED infant TIs and 23% of neonatal intensive care unit (NICU) TIs [67]. VanLooy et al. have established optimal guidelines for neonatal RSI, recommending a sedative and an analgesic as standard, with an NMBA as a third-line option [68]. Their study of 90 neonates demonstrated that adherence to these protocols significantly improved heart rate and oxygen saturation, with 79% of patients requiring three or fewer attempts for successful intubation [68].

Aspiration pneumonitis and pneumonia are key risks in neonatal TI. To reduce the chance of gastric aspiration, cricoid pressure, also known as the “Sellick naneuver”, is often applied during RSI. The recommended force for preventing gastric reflux is between 30 and 40 Newtons (N), but in pediatric patients, the pressure should be reduced due to the higher flexibility of their soft tissues [69]. Pressures exceeding 20 N can cause pain, and proper application of cricoid pressure requires skilled training. It is contraindicated in cases of cricotracheal injury, active vomiting, or unstable cervical spine injuries [69].

##### Endotracheal Tube (ETT) Placement

Hospital-specific approaches to managing difficult neonatal airways are often poorly defined and lack standardization. This is further complicated by the fact that most airway management devices are not specifically designed or tested for use in children [4]. Additionally, many pediatric ETTs are too large for the delicate requirements of neonatal intubation. Despite the critical need, there has been little investigation into appropriate ETT sizing for neonates, and low financial incentives have hindered the mass production of ultrasmall ETTs. Addressing this gap, Peebles et al. conducted a retrospective multicenter cohort study to evaluate the impact of proper ETT sizing on TIAEs [70]. Their findings revealed that using ETTs 0.5 mm smaller than that recommended by the Neonatal Resuscitation Program was associated with significantly lower odds of TIAEs [70]. This suggests that refining tube size recommendations could play a key role in reducing complications during neonatal intubation.

In addition to selecting the appropriate tube size, proper positioning of the ETT is essential for successful intubation. Clinicians generally aim to place the ETT tip in the mid-trachea, and the depth of insertion is often estimated based on the infant’s weight. However, incorrect positioning, particularly over-insertion, is a common issue. To address this, various methods have been proposed to improve placement accuracy. Gill et al. conducted a randomized study of 136 participants from an NICU in Ireland to compare different insertion techniques. They found that neither the vocal cord guide nor a gestation-based formula was superior to the traditional 7-8-9 rule (Tochen’s formula), which estimates the depth of insertion by adding 6 cm to the infant’s birth weight (in kg) [71]. These findings highlight the ongoing challenges in ensuring accurate ETT placement in neonates.

##### Multidisciplinary Approach

NICU patients often require complex, specialized care that spans multiple medical disciplines. While physicians and midlevel providers play a significant role, a team inclusive of respiratory therapists, nurses, nutritionists, and pharmacists is crucial to delivering comprehensive and efficacious care. Ultimately, this is advantageous to the patient, as each discipline offers unique perspectives, expertise, and up-to-date knowledge, which ensures that interventions are evidence-based and tailored to the unique needs of the infant. These benefits are exemplified by a 2024 quality improvement study, which sought to evaluate whether NICU protocols designed by a multidisciplinary team of nurses, respiratory therapists, and physicians could reduce the rate of EUs by at least 10% from baseline. The team developed a formal auditing system to monitor UEs and identify areas of improvement. From monthly audits, the team identified areas of improvement and employed a shared decision-making model to design targeted interventions. At the end of their year-long study, UE rates had substantially improved (a 49% reduction in UE events), demonstrating the importance of multidisciplinary collaboration, especially in the NICU setting [47].

However, implementing such improvements may be challenging in resource-limited settings where NICUs face shortages of trained or specialized staff and infrastructure. In these environments, mitigation strategies can include prioritizing low-cost, high-impact interventions such as simplified protocol standardization, focused in-service staff training, and task shifting. Task shifting is the process of redistributing responsibilities from highly specialized healthcare professionals to other trained personnel with fewer formal qualifications to optimize limited human resources. For example, nurses or general medical staff may be trained to take on specific responsibilities such as ETT securement, airway monitoring, or early identification of UE risks, tasks typically overseen by respiratory therapists or physicians. In addition, checklists, visual guides, and telemedicine support can enhance adherence to safe practices and help maintain quality of care even when specialist availability is limited. While not a full substitute for a multidisciplinary team, these practical adaptations can contribute meaningfully to reducing UE rates in under-resourced NICUs.

The importance of a multidisciplinary approach in the NICU setting is further emphasized by the EXIT (ex utero intrapartum treatment) procedure, which is a specialized surgical technique performed during childbirth to secure the airway of a neonate at risk of obstruction immediately after birth. By allowing for partial delivery while maintaining uteroplacental circulation, this procedure ensures that the neonate continues to receive oxygenated blood from the placenta throughout the intervention [72]. This procedure is typically indicated in cases of an obstructed upper airway, a compromising cervical or thoracic mass, craniofacial anomalies, or if the fetus is suspected to fail the transition to extrauterine life [72,73,74]. The procedure most often occurs in the operating room between 34- and 37-weeks’ gestation and requires coordinated efforts across multiple disciplines to manage the care of both the newborn and the mother. Pediatric otolaryngologists (ENTs) play a significant role in this procedure, as a variety of specialized maneuvers are often required to diagnose the obstruction and secure the neonate’s airway. Some such maneuvers include DL, rigid bronchoscopy with intubation, retrograde intubation, and tracheostomy, among others [73]. However, despite their substantial contribution, especially in cases of difficult airways, few studies have evaluated the impact of ENT involvement on neonatal intubation outcomes.

## 4. Future Perspectives

As the literature demonstrates, neonatal tracheal intubation remains a high-stakes procedure with a substantial risk of adverse events despite advancements in training and equipment. Moving forward, there is a clear need for targeted research aimed at refining current strategies and exploring new interventions that may improve both success rates and patient safety. One promising area is the development and implementation of robust, standardized protocols for managing difficult neonatal airways, which could help mitigate the variability in first-pass success rates and reduce TIAEs. Simulation-based training, especially team-based and high-fidelity modalities, should be further evaluated to assess its effectiveness in preparing healthcare providers for complex airway scenarios.

Additionally, multidisciplinary approaches that incorporate the expertise of neonatologists, anesthesiologists, and otolaryngologists may offer valuable improvements in both procedural success and complication reduction. In particular, the role of otolaryngologists in difficult airway cases remains underexplored in current literature. Future studies should investigate the clinical outcomes, safety metrics, and procedural success rates in settings where otolaryngologists are involved in neonatal intubations. Such research could inform evidence-based guidelines on team composition and escalation protocols during airway emergencies. Overall, advancing neonatal intubation practices requires a collaborative, data-driven approach that prioritizes both technical proficiency and systemic improvements to enhance outcomes for this vulnerable population.

## 5. Conclusions

In summary, neonatal tracheal intubation is a complex and high-risk procedure that requires meticulous preparation, skill, and clinical experience. The literature indicates that while specialized training and the use of advanced equipment can enhance FPSRs, the incidence of TIAEs remains considerable, ranging from 9% to 50% [1]. This wide variability underscores the ongoing challenges in achieving consistent procedural success and minimizing complications such as hypoxemia, bradycardia, BPD, intraventricular hemorrhage, and pneumothorax. Disparities in intubation outcomes across different clinical settings further emphasize the importance of context-specific practices and highlight areas where improvements are still needed. Collectively, the literature supports the necessity for continuous evaluation of current techniques, training protocols, and clinical team structures to improve safety and effectiveness in neonatal airway management.

## Figures and Tables

**Figure 1 healthcare-13-01242-f001:**
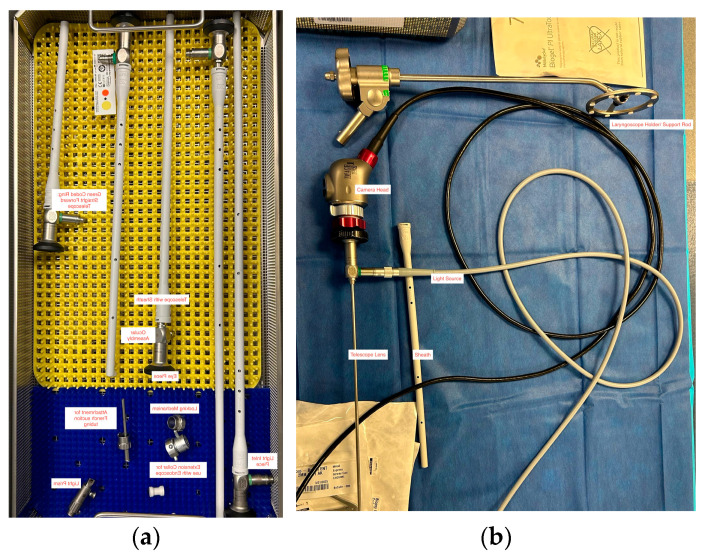
Examples of tools used in neonatal tracheal intubation. (**a**) Five different sizes of Hopkins rods, including the lens telescope, ocular and eye pieces, light inlet pieces, and attachments for suction. (**b**) An example of a video laryngoscope camera head, telescope lens, light source, and laryngoscope support rod.

**Figure 2 healthcare-13-01242-f002:**
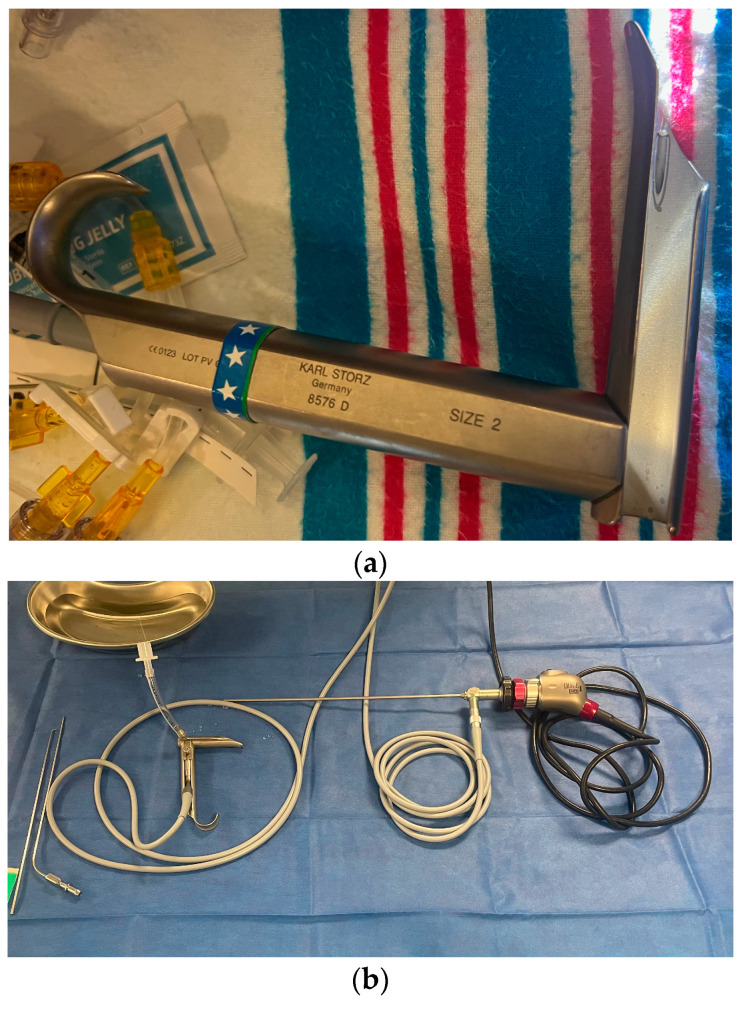
(**a**) A size 2 Parsons laryngoscope, designed for neonates and infants up to 1 year-old. An age-designed laryngoscope is essential for successful pediatric and neonatal intubation. (**b**) The same laryngoscope prepared for use in coordination with operating instruments, video camera access, and anesthetic and oxygen gases.

**Figure 3 healthcare-13-01242-f003:**
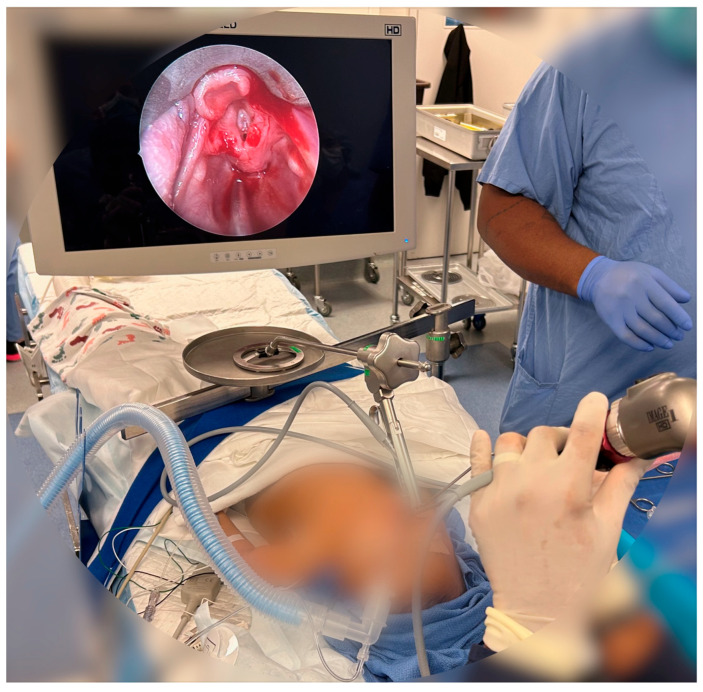
Use of VL in the operating room. VL allows for placement of intubation devices and surgical tools and allows for improved visualization of patient anatomy. The above image shows post-operative viewing of a patient’s larynx using a VL.

**Table 1 healthcare-13-01242-t001:** A summary of common indications for intubation in the NICU setting.

Indication	Description	Evidence
Surfactant Administration	Common among preterm infants with respiratory distress syndrome (RDS)	[1]
Airway Obstruction	Anatomical anomalies like choanal atresia, Pierre Robin sequence, or nasal obstruction	[1,4,5,6]
Elective Airway Management	Planned intubation in stable neonates for controlled procedures or diagnostic purposes	[1,6,7]
Frequent Apnea/Bradycardia Events	Recurrent apneic episodes with bradycardia	[1,8]
Respiratory Failure	The most common indication, including oxygenation and ventilation failure.	[1,8,9]

## Data Availability

No new data were created or analyzed in this study. Data sharing is not applicable to this article.

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
