# Peer review of "Neonatal Tracheal Intubation in the NICU: A Literature Review"

_healthcare, 2025, doi:10.3390/healthcare13111242_

Round 1
Reviewer 1 Report
Comments and Suggestions for Authors
File attached.

Reviewer 2 Report
Comments and Suggestions for Authors
Please see the attached file.

Round 2
Reviewer 1 Report
Comments and Suggestions for Authors
This part of the conclusion is not your study's finding. "Future research should focus on optimizing these strategies, developing robust pro- 556
tocols for managing difficult airways, and evaluating the effectiveness of multidis- 557
ciplinary approaches to neonatal intubation. Additionally, further studies are 558
needed to assess outcomes specifically related to otolaryngologist involvement, par- 559
ticularly in cases of difficult airways, to determine the impact on patient safety and 560
success rates. Improving these aspects is essential to reducing TIAEs and ensuring 561
better outcomes for vulnerable neonates requiring advanced respiratory support. ".
Put a paragraph in the last part of the discussion as a way forward/future perspective, and elaborate in detail.
Author Response
"Put a paragraph in the last part of the discussion as a way forward/future perspective, and elaborate in detail".
To address this feedback, we added separate "Future Perspectives" and "Conclusion" paragraphs.
Reviewer 2 Report
Comments and Suggestions for Authors
The authors have adequately responded to the reviewers' comments and have made meaningful improvements to the manuscript. The revisions have enhanced the clarity, coherence, and overall quality of the work. I am satisfied with the current version and recommend it for acceptance in its present form.
Author Response
Thank you for taking the time to review our manuscript; we sincerely appreciate all feedback you have provided.